# Intranasal Polymeric and Lipid-Based Nanocarriers for CNS Drug Delivery

**DOI:** 10.3390/pharmaceutics15030746

**Published:** 2023-02-23

**Authors:** Rebecca Maher, Almudena Moreno-Borrallo, Dhruvi Jindal, Binh T. Mai, Eduardo Ruiz-Hernandez, Andrew Harkin

**Affiliations:** 1Neuropsychopharmacology Research Group, Trinity College Institute of Neuroscience, Trinity College Dublin, D02 R123 Dublin, Ireland; 2School of Pharmacy and Pharmaceutical Sciences, Trinity College Dublin, D02 YY50 Dublin, Ireland; 3School of Biological and Chemical Sciences, University of Galway, H91 CF50 Galway, Ireland

**Keywords:** central nervous system (CNS), blood–brain barrier (BBB), PLGA nanoparticle (NP), solid lipid NP (SLN), intranasal

## Abstract

Nanomedicine is currently focused on the design and development of nanocarriers that enhance drug delivery to the brain to address unmet clinical needs for treating neuropsychiatric disorders and neurological diseases. Polymer and lipid-based drug carriers are advantageous for delivery to the central nervous system (CNS) due to their safety profiles, drug-loading capacity, and controlled-release properties. Polymer and lipid-based nanoparticles (NPs) are reported to penetrate the blood–brain barrier (BBB) and have been extensively assessed in in vitro and animal models of glioblastoma, epilepsy, and neurodegenerative disease. Since approval by the Food and Drug Administration (FDA) of intranasal esketamine for treatment of major depressive disorder, intranasal administration has emerged as an attractive route to bypass the BBB for drug delivery to the CNS. NPs can be specifically designed for intranasal administration by tailoring their size and coating with mucoadhesive agents or other moieties that promote transport across the nasal mucosa. In this review, unique characteristics of polymeric and lipid-based nanocarriers desirable for drug delivery to the brain are explored in addition to their potential for drug repurposing for the treatment of CNS disorders. Progress in intranasal drug delivery using polymeric and lipid-based nanostructures for the development of treatments of various neurological diseases are also described.

## 1. Introduction

Central nervous system (CNS) diseases, such as Alzheimer’s disease (AD), Parkinson’s disease (PD), epilepsy, brain cancers, and many neuropsychiatric disorders, continue to cause a significant health burden worldwide. However, in the last decade, only 10% of new drugs approved by the US Food and Drug Administration (FDA) are for the treatment of CNS disorders [1]. Although this is a major research field, there is a lack of understanding of the underlying pathophysiological mechanisms of many neurological diseases. This is problematic for drug development and as a result, many available treatments provide only symptomatic relief. Despite promising preclinical evidence, many drug candidates fail in clinical trials due to a lack of efficacy. As an example, this is true for gantenerumab, a human anti-amyloid β (Aβ) monoclonal antibody, which Roche recently announced did not meet the primary endpoints of phase III trials, as amyloid clearance was less than expected and cognitive scores were non-significant (NCT03443973). This, in part, is likely due to poor delivery of the drug to the brain, resulting in a lack of therapeutic effect.

Drug targeting to the brain is a challenge, largely due to the presence of the blood–brain barrier (BBB). This is a highly selective, semipermeable structure comprising many types of brain cells, predominantly vascular endothelial cells, astrocyte end-feet, pericytes, and microglia, which act as gatekeepers to the CNS by controlling the entry of endogenous and exogenous substances (Figure 1; for review of BBB structure and function see Correale and Villa, 2009 [2]). This protective structure also hinders the development of therapies targeting brain diseases as many large or hydrophilic drug molecules are prohibited from crossing (for a recent review of challenges in drug delivery across the BBB, see Harial et al., 2020 [3]). Mechanisms for transiently disrupting the BBB have been proposed, including the use of microbubbles and low-frequency ultrasound waves to temporarily allow larger drug molecules entry [4,5]. However, these processes can render the brain vulnerable to invasion by blood borne toxins or pathogens. Direct drug administration to the brain is highly invasive, can incur serious complications, and is ill-suited for the treatment of long-term diseases that require frequent doses. Due to a growing burden of CNS disorders, there is a demand for novel strategies for the delivery of therapeutics to the brain. 

The field of nanomedicine offers cutting edge solutions to overcome the challenges of drug delivery to the brain. The materials used and the physicochemical properties of nano formulations, such as their shape, size, and surface charge can be tailored according to their purpose [6]. Polymeric and lipid-based nanoparticles (NPs) are emerging as versatile tools for CNS drug delivery, as they are biocompatible and biodegradable, penetrate biological membranes, encapsulate both hydrophobic and hydrophilic drugs, and also provide drug protection and controlled release [7,8]. In particular, poly(lactic co-glycolic acid) (PLGA) NPs and solid lipid nanoparticles (SLNs) have received considerable attention as drug delivery systems to the CNS [9,10]. Drug targeting to the brain can be further enhanced by designing NPs that are coated with molecules that bind specifically to receptors or proteins expressed by the BBB to promote NP uptake by brain cells. 

Additionally, since the recent FDA approval of an intranasal ketamine formulation for the treatment of major depression, nose-to-brain drug delivery has emerged as a promising strategy for bypassing both systemic metabolism and the BBB for improved drug bioavailability in the CNS [11,12]. However, drug uptake via nose-to-brain delivery is also limited by drug properties like molecular weight and lipophilicity [13]. Therefore, an amalgamation of nanomedicine and intranasal drug administration is a promising strategy for drug delivery to the brain. Over the past decade, the growth of the field of both intranasal polymeric and lipid-based nanocarriers has accelerated, reflected by the numbers of publications (Figure 2). Targeted intranasal delivery has the capacity to prevent particle interactions and drug release in the periphery, reducing unwanted side effects, and to deliver the doses required to reach therapeutic concentrations in the brain [14]. Furthermore, nasal sprays or drops are patient-friendly self-administering formulations that do not require a clinical setting, easing the burden of treatment delivery and management on both patient and healthcare provider. 

This review presents the existing evidence that PLGA NPs and SLNs are biocompatible and cross the BBB with the capacity to release cargo to provide therapeutic effects in vitro and in vivo. Furthermore, a growing literature of nose-to-brain delivery is reviewed to assess the potential of intranasal PLGA NP and SLN-mediated drug administration, which aim to improve brain targeting, biodistribution, safety, and efficacy of experimental therapeutics to treat a range of brain disorders. 

## 2. Nanotechnology for BBB Crossing

The recent global introduction of the Moderna and Pfizer-BioNTech COVID-19 lipid NP mRNA vaccines into the clinic has rapidly re-shaped the landscape of nanomedicine, and the number of nano-based therapies entering clinical trials is expected to grow over the coming years [15]. The term “nanomedicine” includes a range of objects at a nanometric scale, such as NPs, nanodrugs, and nanogels [16]. According to their physical nature, NPs can be classified as inorganic or organic [17]. While gold and iron oxide NPs have been approved for clinical applications, particularly in the field of diagnostic radiology [18], the toxicity and clearance of inorganic NPs remains a concern. Several studies have reported an effect on BBB integrity, oxidative stress, and microglial dysfunction [19], therefore making them potentially inadequate as nanocarriers for CNS therapies. Organic nanomaterials have a favourable safety profile and various liposomal, albumin-based, and polymeric nanoformulations are FDA approved for clinical drug delivery [20]. Of these, PLGA NPs and SLNs have emerged as popular candidates for CNS drug delivery. Due to their physicochemical properties, PLGA NPs and SLNs can encapsulate low molecular weight therapies, hydrophilic and hydrophobic drugs, proteins, peptides, vaccine antigens, and gene therapies (Figure 3) [9,21,22,23]. The encapsulated therapeutic agent is protected from metabolism, enzymatic degradation, and premature excretion, thus reducing off-target effects and enhancing drug concentration at the target site.

### 2.1. Polymeric NPs 

Polymeric nanoparticles range in size from 1–999 nm. Synthetic polymers or copolymers of poly(D,L-lactic acid) (PLA), poly(ε-caprolactone) (PCL), PLGA, or natural polymers like chitosan and maltodextrins are used to formulate polymeric NPs. They are synthesised by the self-assembly of two or more chains of block copolymers with varying hydrophobicity using methods like solvent evaporation, nanoprecipitation, super critical fluid technology, and hot or cold homogenisation [16,17]. As PLGA is an FDA approved polymer, PLGA NPs have been extensively studied as drug delivery systems presenting many advantages; they easily cross the BBB, are biocompatible and stable, allow for controlled release kinetics, have high drug loading capacity, and can be functionalized with surface ligands for targeted drug delivery [18,19,20]. Furthermore, PLGA NPs are biodegraded by hydrolysis to produce lactic and glycolic acids, which enter the Kreb’s cycle and are excreted as carbon dioxide and water [21]. Drug release occurs through bulk matrix degradation, however, many environmental factors like pH and the physicochemical NP characteristics can affect the rate of polymer degradation. Therefore, the release pattern is changeable but typically follows a biphasic profile [22]. Increases in PLGA NP size and concentration, as well as changes in shape, have been reported to cause cytotoxicity in vitro, resulting in macrophage activation and the production of reactive oxygen species (ROS). Nevertheless, the body of evidence suggesting that PLGA is biocompatible far exceeds those that describe toxicity, and so, further studies are required to investigate physiological and toxicological responses to PLGA in vivo [23].

### 2.2. Solid Lipid Nanoparticles

SLNs are colloidal nanocarriers that range in size from 50–1000 nm. They are composed of solid physiological lipids, including phospholipids, triglycerides, fatty acids, and steroids, and can be prepared by high pressure homogenisation, ultrasonication/high speed homogenisation, and solvent emulsification/evaporation methods [24]. These preparation techniques have smooth scalability, reproducibility, and the manufacturing process does not involve toxic solvents [25]. Drug incorporation into SLNs can be in the form of a homogenous matrix, a drug-enriched core or a drug enriched shell. Release occurs by particle biodegradation by lipases, erosion, or diffusion, and is dependent on the lipid content, pH, temperature, and the drug entrapment model [26]. Properties of SLNs, such as high surface area and drug loading capacity, controlled release, improved stability, and long-shelf-life make them ideal drug carriers [27]. As SLNs are comprised of biological lipids, they are also biocompatible and easily cross the BBB [28]. Additionally, these lipids have a higher melting point than body temperature and remain in the solid-state post-administration [29,30]. SLNs have been developed and are being tested for many pharmaceutical applications, including the release of anti-tumour drugs like doxorubicin, tamoxifen, docetaxel, and methotrexate; drugs to treat high blood pressure like carvedilol; topical agents like tazarotene used in the treatment of skin conditions; anti-malaria medicine chloroquine; and antitubercular medications like isoniazid and rifampicin [30].

### 2.3. Surface Charge

The surface charge of NPs affects their cellular uptake, biodistribution, and fate in biological systems. Negatively charged NPs present a faster diffusion in tissues and a higher accumulation in tumour tissues when compared to positively charged NPs [31,32]. Due to favourable electrostatic interactions with negatively charged cell membranes, cationic NPs are more easily internalised by cells than neutral or anionic NPs. For this reason, positively charged NPs are more readily taken up by BBB endothelial cell membranes [33]. However, the feasibility of cellular uptake also results in the rapid clearance of cationic NPs from the circulation by macrophages. Additionally, increased liver accumulation is associated with positively charged NPs, which results in prompt plasma clearance and reduced bioavailability [31]. Positively charged NPs may also react with blood components causing haemolysis and toxicity [34]. Furthermore, cationic NPs have been shown to cause cytotoxicity and disrupt the integrity of the BBB, whereas such effects are not reported for neutral and anionic NPs [35]. The surface charge of NPs should be carefully considered in particle design and tailored specifically for the intended purpose. PLGA NPs and SLNs can be positive or negative depending on the synthesis method and may be altered by the surface chemistry.

### 2.4. Surface Modification

Surface engineering of PLGA NPs and SLNs can improve both biocompatibility, brain targeting, stability, and controlled drug release. Polymers like poly(ethylene glycol) (PEG), PCL, chitosan, and PEG-based surfactants like polysorbate 80 and poloxamer 188 can be chemically grafted or adsorbed on the surface of PLGA NPs and SLNs. The hydrophilicity of these moieties increases steric hindrance and circulation time while prohibiting uptake by the reticuloendothelial system (RES) [36]. PEGylation of NPs for CNS drug delivery is common and is reported to improve the circulation time, biocompatibility, and brain uptake, even in pathological conditions [9,10,37,38]. Polymer coatings can also provide drug protection; for example, chitosan modification of SLNs protected against particle degradation at the acidic pH of the stomach following oral administration [39]. 

Proteins, aptamers, peptides, small molecules, and antibodies can also be conjugated to the surface of PLGA NPs and SLNs to improve drug targeting. CNS specific targeting can be achieved using ligands with high affinity for receptors and transporters expressed on the surface of BBB endothelial cells. These ligands include transferrin, lactoferrin, apolipoprotein E, glucose derivatives, and glutathione, which facilitate the brain uptake of NPs through receptor-mediated transcytosis and carrier-mediated transport mechanisms [36]. Cell-penetrating peptides (CPPs) like the transactivator of transcription can also be bound to the surface of NPs through covalent or non-covalent interactions [8]. Conjugation with CPPs can enhance transport through cell membranes, increasing BBB crossing and cellular uptake of drug-loaded NPs [40]. Furthermore, CPPs can overcome the p-glycoprotein (P-gp) efflux pumps expressed by BBB endothelial cells, which are associated with multi-drug resistance [41]. 

Particles can also be conjugated with mucoadhesive agents to facilitate nose-to-brain delivery. Chitosan, a bioactive polymer that improves cell penetration and has mucoadhesive properties, is a commonly used excipient for intranasal drug formulations and can be incorporated into the NP design for nasal delivery (for recent review of chitosan and its mucoadhesive properties, see Aderibigbe et al. (2019) and Mura et al. (2022) [42,43]). Chitosan electrostatically interacts with the negatively charged epithelial surfaces of the nasal cavity to enhance residence time and can also enhance penetration of cell membranes [44]. Additionally, this polymer absorbs water from the mucus lining the nasal cavity, causing the polymer to swell upon contact. This provides a greater surface area for drug crossing through the membrane and into the brain [45,46,47,48]. For this reason, numerous chitosan-based nasal formulations have been proposed as drug delivery systems to the CNS, including chitosan-dopamine and chitosan-tyrosine conjugates for PD [49], chitosan hydrogels for drug delivery in AD [50], chitosan-poloxamer gel for anti-epileptic drug (AED) delivery [51], chitosan nanoemulsions for glioblastoma multiforme (GBM) therapies [52], and chitosan-poloxamer nanoemulsions for the treatment of cerebral ischemia [53]. While NPs can be synthesised from chitosan, it is commonly used as a surface coating to enhance mucoadhesion and particle transport across the nasal mucosa and into the brain. 

### 2.5. PLGA NPs and SLNs Are Compatible with Brain Cells In Vitro

To confirm the safety of PLGA NPs and SLNs in the brain microenvironment, both particle types have been studied in vitro for compatibility with neurons and other resident brain cells. PLGA NPs did not affect the integrity of human SH-SY5Y neuroblastoma cells, monocytes, and 16 HBE epithelial cells used to model the BBB, rodent PC12 catecholaminergic neurons, brain endothelial cells, primary microglia and primary astrocytes, or murine hippocampal neurons, N2a neuroblastoma cells, and N9 microglia [38,54,55,56,57,58,59,60]. Notably, prolonged PLGA NP exposure did not alter neuronal morphology or affect the viability of primary rat neuronal-glial mixed cultures up to concentrations of 2.5 mg/mL [61]. Remarkably, 20 mg/mL PLGA NPs was not toxic to 16HBE cells [62]. Similarly, the application of SLNs to human hCMEC/D3 cerebral vascular endothelial cells, SH-SY5Y cells, primary rodent astrocytes, and brain endothelial cells or mouse BV-2 microglia, brain endothelial cells, and embryonic fibroblasts did not affect cell viability [28,63,64,65,66,67]. 

Furthermore, both PLGA and SLN nanosystems have been deemed compatible with various types of stem cell. The growth of mesenchymal stem cells on PLGA-based platforms was unaffected by the presence of polymeric structures [68]. In a study investigating the potential of SLNs to deliver neuronal differentiation factors to induced pluripotent stem cells (iPSCs), SLNs were non-toxic to stem cells [69]. Flow cytometry revealed no difference in the number of live cells when a human iPSC-based BBB model was exposed to 50 and 100 nm PLGA NPs for 20 h [70], highlighting the potential for the safe translation of these nanocarriers to the clinic for drug delivery to the CNS.

### 2.6. Permeation of In Vitro BBB Models

In vitro models have been established to confirm the ability of PLGA NPs and SLNs to cross the BBB. Cells that make up the BBB can be cultured in a monolayer on transwell devices so that following the application of NPs, the percentage that pass through the cell layer into medium on the basolateral chamber can be quantified (for review of in vitro BBB models, see Williams-Medina et al., 2020 [71]). The modification of PLGA NPs with lactoferrin or anti-transferrin receptor monoclonal antibody increased BBB crossing in vitro [57,72]. Similarly, SLNs effectively crossed cerebral vascular endothelial cells and conjugation with apolipoprotein E or transferrin significantly increased cell uptake [28,63]. In a multicellular BBB model consisting of primary rat brain endothelial cells, astrocytes, and pericytes, SLNs penetrated the barrier and targeting was increased over 3-fold by surface modification with apolipoprotein E [66]. 

### 2.7. PLGA NP and SLN Drug Delivery to In Vitro CNS Disease Models

Prior to in vivo evaluation, PLGA NP and SLN drug delivery vehicles have been evaluated in in vitro models of neuroinflammation, neurodegeneration, and brain cancers to assess drug release and drug action. 

#### 2.7.1. Neurodegenerative Disease

In vitro models of neurodegeneration can be achieved by applying disease salient factors to brain-derived cells. Insights into the in vivo efficacy and therapeutic doses of substances released from PLGA NPs and SLNs can be gained through in vitro screening. PLGA-PEG NP delivery of fucoxantin, a marine carotenoid that is reported to have neuroprotective effects, prevented Aβ-induced neurotoxicity, ROS production, and the release of pro-inflammatory cytokines in SH-SY5Y and BV-2 microglia cells [73]. Pre-treatment with resveratrol-loaded PLGA NPs inhibited H_2_O_2_–induced ROS production and was protective against 1-methyl-4-phenylpyridinium (MPP^+^)-induced mitochondrial dysfunction and cytotoxicity in SH-SY5Y cells as an in vitro model of PD [57]. Similarly, the concurrent application of drug-loaded SLNs with 6-hydroxydopamine (6-OHDA)-induction of an SH-SY5Y cell model of PD was cytoprotective [64]. SLNs also successfully delivered anti-inflammatory therapies to lipopolysaccharide (LPS)-stimulated microglial cells, attenuating nitric oxide production, the expression of nitric oxide synthase and cyclooxygenase-2 (COX-2), and the production of pro-inflammatory cytokines [65]. The release of idebenone, an anti-oxidant agent, from SLNs was protective against 2,2′-azobis-(2-amidinopropane)dihydrochloride-induced oxidative stress in primary rat astrocytes, as measured by a reduction in cytotoxicity and the production of ROS [67]. 

#### 2.7.2. Brain Cancer

Robust in vitro models of brain cancers exist, which involve culturing tumour cells and testing drug efficacy by measuring cell death. PLGA NPs loaded with a derivative of the anti-cancer drug temozolomide were non-toxic to 16HBE cells but reduced the viability of T98G GBM cells to 20% of control [58]. Doxorubicin-entrapped SLNs induced cell death when applied to U87MG GBM cells [74]. Furthermore, PLGA NPs conjugated with an anti-epidermal growth factor receptor (EGFR) monoclonal antibody and loaded with curcumin achieved a reduction in the growth of EGFR-expressing GBM cells at lower concentrations than those required for free curcumin or unmodified curcumin-loaded PLGA NPs to achieve this effect [75]. Lipid-based and polymeric NPs are also being explored for the delivery of chemotherapeutic agents in paediatric cancers (for review see Guido et al., 2022 [76]).

## 3. PLGA NP and SLN-Mediated Drug Delivery In Vivo

While in vitro assessments of drug release and targeting are crucial, CNS disorders are multifactorial, complex diseases involving many physiological processes. For example, epilepsy, neurodevelopmental disorders, and neuropsychiatric disease are difficult to model in vitro. For this reason, it is important to examine both the ability of PLGA NPs and SLNs to successfully deliver therapeutics to the brain and the resulting effects on disease symptoms and pathology in animal models of neurological and neuropsychiatric disease. Currently, there are limited examples of intranasal drug delivery in CNS disease models (see Section 5.2.), and so, evidence of the systemic biocompatibility of PLGA NPs and SLNs, as well as the efficacy of drug delivery following oral, intravenous (i.v.) and intraperitoneal (i.p.) administration, are highlighted here.

### 3.1. Biocompatibility and Brain Distribution of PLGA NPs and SLNs In Vivo

The safety profile and accumulation of PLGA NPs and SLNs in peripheral organs and tissues via various routes of administration have been investigated. A systematic review investigating the safety of SLNs in vivo by oral, parenteral, intranasal, and intratracheal administration reported few occurrences of adverse effects. Those mentioned included microglial activation, neurovascular injury, and liver lesions, but these were attributed to factors like surface charge, NP aggregation, irritation of administration site, and a large drug load encapsulated within the NPs [77]. In an evaluation of the long-term effect of thrice weekly i.p. administration of high concentrations of PLGA NPs and SLNs (100 mg/kg), no mortality was reported, and the treatment did not affect body weight [78]. This is in agreement with Casanova et al. (2022), who reported that i.p. administration of PLGA NPs every 3 days for 43 days had no effect on the body weight of rats [79]. The harvesting and analysis of major organs following prolonged exposure to PLGA NPs and SLNs revealed some accumulation of particles in the liver, spleen, and bone marrow (only in females), but not in the lungs, heart, or kidneys [78]. Despite the accumulation of particles, no significant histopathological differences were found in the liver, spleen, and bone marrow, along with no signs of toxicity in the brain, heart, spleen, and thymus of i.p. PLGA NP or SLN-treated mice [78]. 

Fluorescently labelled NPs can be tracked as they move throughout the body. In a mouse model of traumatic brain injury (TBI), a biocompatible cyanine dye was used as both a targeting agent to the site of necrosis and as a means of localising the distribution of PEGylated PLGA NPs (100, 200, and 800 nm) following i.v. administration. Whole-body fluorescent imaging and histological analysis determined that smaller PLGA NPs were more effectively targeted to the brain over a 48-h period and, more specifically, had higher accumulation at the lesioned area than larger particles. However, all sizes of PLGA NPs were detected in the liver after 48 h, where an increase in size was associated with higher liver retention [80]. This is supported by a systematic review that reports a decrease in PLGA NP cytotoxicity directly relating to a decrease in particle size [81]. However, the PLGA NP formulations that were highlighted as toxic in this analysis were loaded with anti-cancer agents, which are known to be damaging to healthy cells, or non-biocompatible magnetic agents for imaging [81]. Albumin-coated PLGA NPs were traced and located throughout the brain, where notably, the striatum had the highest accumulation out of all brain regions when a high concentration of 20 mg/mL was administered i.p. [82]. Fluorescent SLNs can enter the brain parenchyma as early as 3 h and up to 72 h following i.v. administration in rats [83]. Furthermore, pharmacokinetic analyses have confirmed the ability of PLGA NPs and SLNs to release cargo into the brain tissue when administered i.v., i.p., and orally in rodents [84,85,86,87,88,89,90].

### 3.2. PLGA NPs and SLNs as Drug Delivery Vehicles for CNS Disease: Preclinical Evidence

The efficacy of drugs delivered by PLGA NPs and SLNs has been studied in animal models of CNS disease. This includes currently used drugs that may benefit from nanocarrier formulations to reduce the required dose or improve the side effect profile, as well as compounds whose therapeutic potential has not been exploited due to poor bioavailability or insufficient brain uptake. The physicochemical properties of the PLGA NPs and SLNs described here can be found in Table 1. 

FDA approved drugs for CNS diseases can have low bioavailability and poor penetration of the BBB. Preclinical evidence has demonstrated that the encapsulation of such drugs in PLGA NPs and SLNs improves both the pharmacokinetic profile and drug efficacy in animal models when compared to free drug formulations. These include galantamine for the treatment of AD [91]; anti-PD therapies tolcapone, and resveratrol [57,79]; chemotherapies like paclitaxel [92]; AEDs carbamazepine, and levetiracetam [93]; anti-depressant drug duloxetine [94]; atypical antipsychotic lurasidone hydrochloride [95]; and naloxone, which is used in the treatment of substance abuse disorder [96]. 

Many natural products are reported to have antioxidant, neuroprotective, and anti-inflammatory effects, but due to instability in biological fluids, rapid metabolism, and inability to cross the BBB, they have yet to be harnessed as pharmacotherapies for the treatment of CNS diseases. The encapsulation of natural products like phytol, epigallocatechin-3-gallate (EGCG), nicotinamide, and curcumin in PLGA NPs and SLNs improved the bioavailability of these compounds compared to drug solutions. Furthermore, in vivo pharmacodynamic studies have demonstrated an improved efficacy of nanoformulations of natural products in AD, PD, and Huntington’s disease (HD) [55,97,98,99,100,101].

A depletion in biological molecules can occur as a result of or contribute to the pathophysiology of CNS disease. However, the delivery of such biologics to the brain is impeded by peripheral metabolism and an inability to cross the BBB. PLGA NP and SLN-mediated brain delivery of neuro-signalling molecules like dopamine, glycoproteins like vitamin D-binding protein, and lipids like cholesterol, enhanced the brain concentrations of these biologics and produced therapeutic effects in animal models of PD [82], AD [59], and HD [102], respectively. 

The drug development process is long, arduous, and expensive from initial drug discovery to the clinical trial phase, and even then, FDA approval is certainly not guaranteed. Nanomedicine has the potential to facilitate drug repurposing, where currently approved medicines that do not readily cross the BBB can be targeted to the brain and deliver therapeutics for CNS disease. An example of this approach is pioglitazone, a peroxisome proliferator-activated receptor γ (PPARγ) agonist that is currently used to treat diabetes but is also reported to have neuroprotective properties. The encapsulation of pioglitazone in PLGA-PEG NPs and oral delivery reduced memory deficits and decreased Aβ load in the APP/PS1 mouse model of AD compared to the bulk drug [103]. 

CNS diseases tend to have a large genetic component and novel technologies can revolutionise gene therapies for brain disorders. However, siRNA is unstable in biological fluids and is also negatively charged, inhibiting the penetration of the anionic cell membranes of the BBB. Therefore, the development of gene therapies for CNS disease has been slow to progress and may benefit from NP delivery systems. PLGA NPs delivered siRNA to a mouse model of GBM, and specifically silenced tumour genes related to cell proliferation, resulting in reduced tumour volume [104].

**Table 1 pharmaceutics-15-00746-t001:** Physicochemical properties of PLGA NPs and SLNs delivered in vivo.

Disease	Nanocarrier	Disease Model	Drug Loaded	Mode of Action	Size (nm)	PDI	Zeta Potential (mV)	ROA	Drug Conc. Administered by NP	Outcomes	Ref.
Alzheimer’s disease											
	SLN	Isoproterenol-induced rat model	Galantamine	AChE inhibitor	88 ± 1.89–221.4 ± 1.34	0.275 ± 0.12–0.380 ±0.16	−10.04 ± 1.9–−18.75 ± 1.7	Oral	5 mg/kg for 3 weeks	Galantamine-loaded SLNs protected against memory impairments	[91]
	PLGA NP	Scopolamine-induced rat model	Phytol	Antioxidant, anti-inflammatory, anti-amyloid	177.4 ± 5.9	0.2 ± 0.06	−32.8 ± 2.2.	Oral	100 mg/kg or 200 mg/kg	Improved spatial & short memory, prevented acetylcholine breakdown and regulated neuronal death, reduced oxidative stress	[101,105]
	PLGA-PEG NP	Transgenic mouse model	Pioglitazone	Neuroprotection	155 ± 1.8	0.1	−13 ± 0.5	Oral	10 mg/kg; 5 days a week for 4 weeks	Reduced memory impairment and fewer cortical Aβ deposits	[103]
	PLGA-PEG NP	Transgenic mouse model	ECGC	Antioxidant, neuroprotection	124.8 ± 5.2	0.054 ± 0.013	−15	Oral	40 mg/kg daily for 3 months	Improved spatial learning and memory, increased number of synapses, reduced neuroinflammation and Aβ burden	[100]
	PLGA NP	Transgenic mouse model	Vitamin D-binding protein	Anti-amyloid	226.6 ± 44.4	0.039 ± 0.013	−0.144	i.v.	2.5 mg/kg of NPs daily for 4 weeks	Inhibited Aβ aggregation, neuroinflammation, neuronal death and cognitive deficits	[59]
	SLN	Streptozotocin rat model	Nicotinamide	Cognitive enhancer	124 ± 0.8	0.831	−12.5 ± 0.7	i.p. and i.v.	60, 30, 15 mg/kg every other day	Improved cognition, neuroprotection and reduced tau hyperphosphorylation	[99]
Vascular Dementia											
	SLN	Homocysteine rat model	Curcumin	Antioxidant, anti-inflammatory	154.8	0.928	−10.9	Oral	25 mg/kg daily for 2 weeks	Improved memory, reduced oxidative stress biomarkers, reduced AChE activity, increased GABA, decreased glutamate and exerted neuroprotection in the cortex and hippocampus	[98]
Parkinson’s disease											
	Lactoferrin-PLGA NP	MPTP mouse model	Resveratrol	Antioxidant, anti-inflammatory, neuroprotective	148.2 ± 4.2	0.12 ± 0.18	−23.1 ± 3.0	i.v.	5 mg/kg every other day for 15 days	Improved motor functions, protected against DA depletion, neuroprotective and reduced glial activation and neuroinflammation in the SN	[57]
	PLGA NP	Rotenone rat model	Tolcapone	Reduces dopamine metabolism	182.59 ± 23.94	Not stated	−26.32 ± 0.48	i.p.	3 mg/kg every 3 days for 45 days	Improved motor functions, prevented nigral cell death, reduced glial activation	[79]
	Albumin-PLGA NP	6-OHDA mouse model	Dopamine	Dopamine replenishment	353	0.5	−37	i.p.	0.05 mg/μL or 0.1 mg/μL	Increased dopamine, improved motor coordination, balance and sensorimotor functions	[82]
	SLN	Rotenone mouse model	Curcumin	Antioxidant, anti-inflammatory	134.5 ± 0.85	0.39 ±0.04	−18.56 ± 0.55	Transdermal	85 mg/kg	Decreased bradykinesia, improved coordination and balance	[97]
Huntington’s disease											
	PLGA-PEG NP	3-nitropropionic acid mouse model	EGCG	Antioxidant, neuroprotection	124.8 ± 5.2	0.054 ± 0.013	−15.7 ± 1.7	i.p.	50 mg/kg daily for 5 days	Relieved motor symptoms, neuroprotective and reduced neuroinflammation	[55]
	Glyco-protein7 -PLGA NP	Transgenic mouse model	Cholesterol	Slows disease progression	249 ± 38	0.29 ± 0.05	−30 ± 7	i.p.	1.7 mg NPs/mouse twice weekly for 5 weeks	Delayed onset of symptoms in pre-symptomatic mice, rescued cognitive decline in symptomatic mice, improved motor recovery, reduced muHTT aggregation	[102]
GBM											
	SPIO-PLGA NP	Orthotopic U87MG tumour mouse model	Paclitaxel	Prevents cancer cell growth and induces cell death	250 ± 20	0.11	−18 ± 5	i.v.	5 mg/kg every 4 days for 16 days starting 8 days post tumour inoculation	Improved survival time	[92]
	PLGA-PEG NP	Orthotopic U87MG tumour mouse model	siRNA targeting hepatocyte growth factor receptor	Reduces tumour cell proliferation	117.4 ± 11.7	Not stated	37.3 ± 2.3	i.v.	0.125, 0.5 or 2 mg/kg three times a week for 3 weeks, two weeks post tumour inoculation	Reduced tumour volume	[104]
Epilepsy											
	PLGA NP	Pentylene- tetrazole induced seizure rat model	Carbamaze-pine and levetiracetam	Reduces epileptic activity	180.62 ± 6.26	0.107 ± 0.03	−27.08 ± 3.11	i.p.	30 mg/kg carbamazepine & 1.2 mg/kg levetiracetam	Decreased seizure activity	[93]
Depression											
	SLN	LPS rat model	Duloxetine	Reduces symptoms of depression	114.5 ± 2	0.29 ± 0.03	−18.2 ± 1.8	i.p.	30 mg/kg daily for 14 days	Decreased immobility time, reduced TNFα and COX-2 expression	[94]
Schizophrenia											
	SLN	Dizocilpine rat model	Lurasidone hydrochloride	Reduces symptoms of psychosis	139.8 ± 5.5.	0.118 ± 0.002	−30.8 ± 3.5	oral	2.066 mg/kg for 3 weeks	Improved cognition and reduced EPS effects	[95]
Substance abuse disorder											
	PLGA NP	Fentanyl-dependent rat	Naloxone	Opioid receptor antagonist	263	0.2	Not stated	i.m.	10 mg/kg	Prevented fentanyl induced antinociception and respiratory depression	[106]

PDI; polydisperity index, ROA; route of administration, SLN; solid lipid nanoparticle, PLGA NP; poly(lactic-co-glycolic acid) nanoparticle, PEG; polyethylene glycol, GBM; glioblastoma multiforme, AChE; acetylcholinesterase, Aβ; amyloid β, ECGC; epigallocatechin-3-gallate; DA; dopamine, MPTP; 1-methyl-4-phenyl-1,2,3,6-tetrahydropyridine, 6-OHDA; 6-hydroxydopamine, L-Dopa; levodopa, LPS; lipopolysaccharide, TNFα; tumour necrosis factor α; COX-2; cyclooxygenase-2, EPS; extrapyramidal symptoms, i.v.; intravenous, i.p.; intraperitoneal, i.m.; intramuscular.

## 4. Intranasal Drug Delivery

Indeed, nanomedicine offers promising solutions for the delivery of therapeutics to the brain. However, when delivered systemically, NPs are subjected to similar obstacles faced by drugs in solution. NPs may be metabolised in the periphery, releasing the drug before it gets to the brain or become immobilised in organs or tissues, hindering CNS bioavailability. Recently, the intranasal route of administration has emerged as a promising approach to enhance the brain uptake of drug molecules or biological agents by bypassing the BBB entirely. Drugs are primarily transported from the nasal cavity to the brain parenchyma by direct transport along the olfactory nerve or indirectly via paracellular or transcellular transport across the nasal epithelium and eventually reach the brain by the blood or cerebrospinal fluid (Figure 4; for recent review of nose-to-brain drug delivery see Crowe & Hsu, 2022 [107]). A combination of these pathways likely contributes to drug transport from the nasal cavity to the brain, depending on the properties of the drug formulation. 

In 2019, an intranasal formulation of esketamine SPRAVATO^®^ was approved by the FDA and European Medical Agency (EMA) as a fast-acting antidepressant for the treatment of major depressive disorder (MDD) following positive results from clinical trials, including rapid relief of depressive symptoms like suicidal ideation, improved mood, long-lasting effects, and a favourable safety profile [109,110,111,112]. Following this, the potential use of intranasal esketamine for bipolar disorder and MDD with comorbidities, such as post-traumatic stress disorder and psychosis, has been explored in patient populations with similar positive outcomes [113,114,115]. The approval of intranasal esketamine for clinical use has accelerated research into this administration route for CNS drug delivery, particularly in cases where a pharmacotherapy is rapidly metabolised in the periphery, cannot easily penetrate the BBB or is associated with systemic adverse effects.

### 4.1. Nasal Drug Delivery Bypasses the BBB

Bypassing the BBB is a major obstacle when delivering peptide therapeutics to the CNS. Oxytocin, a peptide hormone produced in the hypothalamus, has been shown to reverse the effects of Aβ on long-term potentiation ex vivo [116]. However, when administered orally, oxytocin is rapidly metabolised by the liver and kidneys, so a targeted approach is required to achieve a therapeutic effect in the brain. Takahasai et al. (2022) reported a comparable pharmacological profile for similar doses of oxytocin delivered intracerebroventricularly (i.c.v) and intranasally for the treatment of Aβ-induced memory impairment in mice. Furthermore, intranasal oxytocin attenuated the cognitive deficits caused by Aβ [117]. Oxytocin has also been reported to alleviate the core symptoms of autism spectrum disorder (ASD) and when administered intranasally to adolescent animals that were prenatally exposed to valproic acid, ASD-like phenotypes were ameliorated at a gene expression level [118]. Intranasal oxytocin is also reported to have antipsychotic effects and has been trialled as an adjunctive therapy for schizophrenia. The results of clinical trials are inconsistent, and it is likely that high doses are required to achieve a therapeutic effect, however, further investigation is necessary to make a meaningful conclusion [119]. Similarly, intranasal neuropeptide Y and galanin receptor agonists improved performance in memory retrieval tasks and increased cell proliferation in the dorsal hippocampus of rats [120]. 

Glial cell-derived neurotrophic factor (GDNF) is neuroprotective and has been implicated in the growth and repair of dopamine neurons in the substantia nigra [121,122]. As GDNF does not easily cross the BBB, the intranasal route has been explored in the 6-OHDA model of PD. Rats treated with intranasal GDNF had a higher number of tyrosine hydroxylase-positive dopamine neurons compared to control lesioned rats, indicating a protective effect of intranasal GDNF [123]. Likewise, nerve-growth factor (NGF) supports neuronal growth and repair and has been shown to be neuroprotective upon intraventricular administration following brain injury, both experimentally and in small patient cohorts [124,125]. In a case report of a four-year-old boy in a persistent unresponsive wakefulness syndrome, intranasal NGF improved functional positron emission tomography (PET), computed tomography (CT), and magnetic resonance imaging (MRI) outcomes as well as voluntary movements, attention, verbal comprehension, and bowel and urinary function [126]. 

The development of immunotherapies and gene therapies for substance abuse disorder is a novel approach to treatment. Thus far, two anti-cocaine vaccines have entered clinical trials; anti-cocaine vaccine TA-CD failed due to a lack of efficacy in phase III trials [127] and dAd5GNE, an adenovirus gene therapy conjugated to a cocaine analogue, is currently undergoing phase I evaluations [128]. To overcome the challenges in transporting such therapies across the BBB, Lin et al. (2022) have developed an intranasal immunization against cocaine using a synthetic polymer as an adjuvant. The intranasal administration of the anti-cocaine vaccine attenuated cocaine induced locomotor activity, produced a robust IgG and IgA response in mice, and had comparable efficacy to i.p. administration [129]. 

The intranasal route is also considered a favourable strategy for the delivery of stem cells to the brain, which typically require an invasive direct application. Reitz et al. (2012) delivered neural progenitor cells intranasally to GBM-bearing mice which were successfully targeted to peri- and intra-tumour regions as early as 6 h post intranasal administration [130]. 

Studies have identified efflux transporter proteins, such as P-gp and breast cancer related protein (BCRP), which are expressed by brain capillary endothelial cells of the BBB, as a critical contributor to AED resistance in the treatment of epilepsy [131]. Therefore, nose-to-brain administration is a promising solution to overcome drug resistance by bypassing the BBB entirely. The i.v. administration of zonisamide in the presence a BCRP inhibitor resulted in higher drug accumulation in the brain than in the absence of an inhibitor. However, brain concentrations of a nasal zonisamide formulation were unaffected by BCRP-inhibition, indicating that nose-to-brain delivery circumvents the action of BBB transporters responsible for resistance to CNS drugs [132]. 

Intranasal drug administration may enable the repurposing of therapeutics that are FDA approved for alternative uses that do not readily cross the BBB. Insulin, a hormone secreted by the pancreas, is commonly used to treat type I diabetes and is dysregulated in several neurodegenerative and neuropsychiatric disorders [133]. The intranasal route is under investigation as systemically administered insulin can be deployed for glucose storage in the periphery in addition to reaching the CNS. Insulin was shown to improve locomotor activity and prevent dopaminergic neuronal loss when delivered intranasally pre- and post-lesion in the 6-OHDA rat model of PD [134,135]. Intranasal insulin has now reached clinical trials for PD, with preliminary results reporting both safety and functional improvements [136]. 

### 4.2. Intranasal Formulations Reduce Side Effects

A considerable challenge in the long-term management of epilepsy is the serious side effects that patients experience, especially for those prescribed a cocktail of AEDs. Not only does this affect a patient’s quality of life, but it also reduces therapeutic adherence. Nasal spray formulations of commonly used AEDs, such as diazepam and midazolam, have recently been FDA-approved for clinical use [137,138]. Data collected during various clinical trials suggests that both intranasal formulations have improved patient quality of life, are safe for use by children and adolescents, and they are effective at the cessation of seizure clusters and of acute seizures compared to alternative administration routes [139,140,141,142]. Novel treatments for epilepsy are also being formulated for nasal administration; siRNA was effectively used to silence the GluN1 gene, which encodes the GluN1 subunit of the AMPA receptor, in the hippocampus to reduce excitatory neurotransmission and epileptic activity. Intranasal administration of this siRNA in the pilocarpine model of temporal-lobe epilepsy significantly increased the latency time for the animals first seizure [143]. 

Chemotherapy notoriously causes a plethora of systemic side effects, and the treatment of brain cancers would benefit greatly from nose-to-brain delivery. An intranasal chemotherapy formulation of perillyl alcohol, which has been shown to have chemotherapeutic effects, is currently in phase I/II clinical trials for GBM, with results expected in 2024 (NCT02704858) [144].

Teriflunomide is used clinically for the treatment of multiple sclerosis; however, systemic administration is associated with serious liver damage. Studies have shown that teriflunomide has anti-cancer properties, which led Gadhave et al. (2019) to investigate intranasal administration of this drug in rats. Preliminary pharmacokinetic analyses revealed that teriflunomide accumulation in the brain was two-fold higher following intranasal administration compared to i.v. administration, and no changes in liver biomarkers, haematology, or histopathology were reported [145]. 

### 4.3. Strategies for Improving Nose-to-Brain Transport 

The olfactory pathway is the predominant route to the brain, meaning optimal drug penetration occurs when a drug or drug carrier adheres to the olfactory region of the nasal cavity. Mucociliary clearance occurs every 15–20 min, so rapid absorption across the mucosal membrane is a necessary feature of nasal formulations. Drug uptake is most effective when therapeutics are formulated to enhance the residence time in the nasal cavity and promote drug penetration. 

Microemulsions and gels have been formulated for the purpose of intranasal drug delivery. Recent preclinical advances in this field include the development of thermo-, pH-, and ion-sensitive hydrogels and polymeric gels, microemulsions, and nanoemulsions for intranasal drug delivery in AD, PD, epilepsy, GBM, depression, schizophrenia, and sleep disorders [50,146,147,148,149,150,151,152,153,154,155,156,157]. Intranasal delivery of these drug carriers improved nose-to-brain delivery and the safety profile of the drug compared to solution and enhanced drug efficacy in animal models of CNS disease. 

## 5. Intranasal Delivery of Experimental Therapeutics to the CNS via PLGA NPs and SLNs

While systemic administration of drug-loaded PLGA NPs and SLNs is safe and has disease-modifying results in animal models of CNS disease, oral NPs are subjected to clearance or tissue binding leading to poor distribution. Following systemic administration, PLGA NPs and SLNs are rapidly cleared by the RES, causing accumulation in related organs (liver, spleen, lung, and kidneys) [158]. Moreover, orally delivered SLNs can be degraded by lipases in the gut [158]. Peripheral metabolism of NPs significantly affects brain bioavailability, which may be negated by direct targeting to the brain via the intranasal route. Differences in the accumulation of particles in different organs between males and female rodents may lead to sex-dependent side effects in the clinic, which may also be avoided by direct CNS targeting. Furthermore, the limitations associated with intranasal drug delivery, such as low dosage volumes, nasal mucosa impenetrability of high molecular weight drugs, mucociliary clearance, and enzymatic drug degradation [13], may be overcome using nanoparticle formulations as drug carriers for nose-to-brain delivery. 

A combination of PLGA and SLN delivery systems with intranasal delivery has the potential to unlock novel therapeutic strategies for CNS disease, particularly for compounds that have poor brain uptake, associated peripheral toxicities, or adverse effects that are unsuitable for long-term therapies.

### 5.1. Brain Distribution and Drug Bioavailability of Intranasal PLGA NPs and SLNs

Preliminary pharmacokinetic analyses have tested the ability of intranasally administered PLGA NPs and SLNs to deliver experimental therapeutics to the brain compared to drug solutions and alternative administration routes in several CNS diseases. This includes drugs that are currently in use but may benefit from intranasal NP delivery systems to improve their side effect profile, as well as drugs whose clinical potential is yet to be unlocked due to poor BBB permeability or associated toxicities. 

Intranasal PLGA NPs and SLNs have been shown to deliver therapeutics to the brain in concentrations higher than alternative administration routes including oral, i.p., and i.v. and formulations including drug in solution or bulk drug. This is true for FDA approved therapies like L-Dopa, paclitaxel, carmustine, lamotrigine, carbamazepine, desvenlafaxine, almotriptan, naloxone, experimental immunotherapies, and natural products, including ferulic acid (FA), isoflavonoids, and catechins [159,160,161,162,163,164,165,166,167,168,169,170]. Brain targeting is further improved by NP surface modifications, such as chitosan, lactoferrin, transferrin, and PEG coatings, which enhance the transport of NPs from the nose to the brain [62,165,168,170,171,172].

Furthermore, toxicological analyses in rodents do not report evidence of significant levels of PLGA or SLNs in major organs or tissue damage following intranasal administration, and body weight did not fluctuate dramatically throughout treatments [160,164,171,172,173,174]. In support of this, there is no evidence of fatalities associated with intranasal PLGA NP or SLN treatment. One study detected uncoated PLGA NPs in the lungs, likely due to inhalation of the particles, highlighting the importance of mucoadhesive formulations for absorption across the nasal mucosa [62]. 

Drugs like anti-amyloid antibody gantenerumab, potassium channel activator Maxipost, anti-inflammatory celecoxib, and antibiotic minocycline, that were successful preclinically but failed in clinical trials for AD [175], stroke [176], and depression [177], respectively, may benefit from re-evaluation in a nose-to-brain nanosystem. While agomelatine is EMA approved for the treatment of MDD, the FDA did not approve agomelatine due to reports of hepatotoxicity [178]. Furthermore, agomelatine undergoes substantial first-pass metabolism, resulting in less than 5% bioavailability when administered orally. The intranasal delivery of agomelatine-loaded SLNs to rats resulted in higher brain concentrations of drug compared to i.v. agomelatine and the commercially available oral formulation Valdoxan^®^ [179]. 

Nose-to-brain delivery of drug loaded NPs may enable drug repurposing. Novel therapeutic strategies are desperately needed for pain management, as often patients can become reliant on opioids and are at risk of dependence. In some cases of neuropathic pain, AEDs like lamotrigine are prescribed to reduce neuronal excitability. However, following oral administration, brain bioavailability is low. The brain targeting of lamotrigine was improved by encapsulation in PLGA NPs and intranasal delivery compared to i.v. lamotrigine-loaded PLGA NPs and i.v. aqueous lamotrigine [180].

### 5.2. Proof of Concept: Efficacy of Drug-Loaded Intranasal PLGA NPs and SLNs in Animal Models of CNS Disease

While the available evidence suggests that the pharmacokinetic profile of CNS therapies can be improved with intranasal polymeric and lipid nanoformulations, there is a relative paucity of research on their effects in animal models of CNS disease (Table 2). 

To date, the majority of pharmacodynamic studies have investigated PLGA NPs and SLNs as intranasal carriers of FDA-approved CNS therapies. Having confirmed a higher drug concentration in the brain following intranasal delivery of drug loaded PLGA NPs and SLNs compared to oral and i.v. routes, Kaur et al. (2022) confirmed that pre-treatment with FDA approved N-methyl-D-aspartate antagonist memantine-loaded PLGA NPs delivered intranasally conferred higher protection against scopolamine-induced spatial memory deficits than an intranasal aqueous drug solution in rats [54]. Similarly, when compared to oral L-Dopa administration and intranasal unmodified PLGA-PEG NPs loaded with FDA approved dopamine agonist rotigotine, rotigotine delivered intranasally by lactoferrin-PLGA-PEG NPs enhanced dopaminergic neurotransmission and reduced degeneration while also exhibiting a longer duration of action in the 6-OHDA model of PD [173]. L-Dopa encapsulation in PLGA NPs and nasal administration also improved motor deficits and prolonged drug action compared to intranasal and oral free drug in the MPTP mouse model of PD [174]. The anti-glioma effects of anti-vascular endothelial growth factor immunotherapy bevacizumab, temozolomide derivative TMZ16e, and cell-cycle inhibitor paclitaxel-loaded PLGA NPs were apparent after intranasal administration in mouse models of GBM [58,163,181]. The AED lamotrigine and anti-depressant agent desvenlafaxine were also found to delay seizure onset and reduce symptoms of depression, respectively, when delivered through the nose in PLGA NPs to rats rather than by nasal or i.v. aqueous drug formulations [161,164].

The therapeutic effect of natural products has been known for centuries; however, this class of pharmaceutics remains largely unexploited for CNS drug delivery. Saini et al. (2021) demonstrated that intranasally administrated ferulic acid (FA)-loaded SLNs significantly improved the memory deficits induced by streptozotocin (a diabetogenic compound that induces ROS production and promotes AD pathology), compared to intranasal and oral FA and oral FA-SLN-treated animals [166]. Furthermore, oxidative stress biomarkers and acetylcholinesterase activity were reduced in the cortex and hippocampus of intranasal FA-SLN-treated animals [166]. In a rat model of cerebral ischemia, PLGA NPs were employed for the nose-to-brain delivery of glycosyloxyflavone baicalin and successfully attenuated neuroinflammation [182]. In the pentylenetetrazole- and electroshock-induced seizure model, nasal delivery of phytochemical catechin hydrate-loaded PLGA NPs conjugated with chitosan also had anti-seizure effects in rats [170]. 

Evidently, PLGA NP and SLN-mediated nose-to-brain delivery not only enables higher drug concentrations to reach the brain but also maintains or enhances the therapeutic effect of several FDA approved drugs in animal models of CNS disease. While there are limitations of intranasal drug administration, including mucociliary clearance, a small absorption area, and the possibility of inhalation and accumulation in the lungs, specifically designed nanoparticulate systems can be deployed to overcome these. Based on the evidence presented here, for the successful delivery of CNS drugs to the brain, the following should be considered in the design of intranasal NPs; the use of biocompatible and lipophilic materials for NP synthesis, particle size, surface charge, and mucoadhesion. Furthermore, this combinatory drug delivery approach unlocks the possibility of novel therapeutic agents for treating CNS disease.

### 5.3. Concerns Regarding Intranasal Delivery of Nanomedicine

The nasal mucosa is a first line defence mechanism of the innate immune system. Therefore, any foreign object that enters the nasal cavity has the propensity to cause an immune response. Nasal spray bottles may harbour bacteria and so, special attention is required by patients or carers to prevent this from causing an infection [183]. The surface area of the nasal cavity is about 12 cm long with the olfactory mucosa accounting for only about 16 mm of this [184,185]. Therefore, a small absorption area is a concern for nasal administration, and it is difficult to ensure that the correct dose is transported from nose-to-brain. However, nasal spray formulations with sufficient force can effectively target the olfactory bulb, and the use of mucoadhesive excipients can improve drug transport to the brain. 

According to a recent systemic review, intranasal formulations of corticosteroids, anti-histamines and alpha adrenergics had the highest incidence of adverse effects, which included dyspnea, headache, epistaxis, and changes in taste and smell [186]. However, these drugs have few actions in the brain and are unlikely to be used in the treatment of CNS disease. Few side effects are reported following intranasal administration of drugs targeted to the CNS. Those mentioned include nasal irritation, itching, and damage to the nasal tissue [187]. In phase III clinical trials, no evidence of adverse effects on olfaction or nasal health were reported following short-term or long-term administration of intranasal esketamine spray [188]. Many of the reported side effects can be overcome by nanomedicine and particle design, including tailoring particle size to avoid irritation and the use of mucoadhesive agents to prevent particles travelling to the airways. 

**Table 2 pharmaceutics-15-00746-t002:** The physicochemical properties of PLGA NPs and SLNs delivered intranasally to animal models of CNS disease.

Disease	Nanocarrier	Disease Model	Drug Loaded	Mode of Action	Size (nm)	PDI	Zeta Potential (mV)	Drug conc. Administered by NP	Outcomes	Ref.
Alzheimer’s disease										
	PLGA NP	Scopolamine rat model	Memantine	NMDA antagonist; cognition enhancer	58.04	0.204	−23	0.1 mg/kg in 20 μL daily for 9 days	Improved spatial memory	[54]
	Chitosan-SLN	Streptozotocin rat model	Ferulic acid	Antioxidant, neuroprotective properties	184.9	0.277	12.4	80 mg/kg for 28 days	Enhanced cognition, reduced oxidative stress and AChE activity in the cortex and hippocampus	[166]
Parkinson’s Disease										
	Lactoferrin PLGA-PEG NP	6-OHDA rat model	Rotigotine	DA agonist; improves dopaminergic neurotransmision	118 ± 12.14	Not stated	−21.94 ± 2.83	2 mg/kg in 200 μL twice daily for 1 week	Improved dopaminergic transmission, reduced nigro-striatal neurodegeneration	[173]
	WGA-PLGA NP	MPTP mouse model	L-Dopa	DA precursor; increases brain levels of DA and transmission	329 ± 188.3	0.384 ± 0.113	−4.47 ± 0.576	16 mg/kg in 20 μL for 7 days	Improved locomotor activity	[174]
GBM										
	PLGA NP	U87 luciferase tumour bearing nude mouse model	Bevacizumab	Anti-VEGF; anti-angiogenesis and tumour cell death	185.0 ± 3.0	0.056 ± 0.016	−2.50 ± 0.27	5 mg/kg in 5 μL weekly for 24 days	Reduced tumour growth and reduced VEGF expression and synthesis	[181]
	Anti-EphA3 PLGA NP	T98G tumour bearing nude mouse model	Temozolomide derivative	Cell cycle arrest; tumour cell death	135.1 ± 2.4	0.085 ± 0.037	−28.65 ± 1.2	5 mg/kg when tumour reached 5 mm for 15 days	Increased survival time, increased apoptosis of tumour cells	[58]
	PLGA NP	U87MG tumour bearing mouse model	Paclitaxel	Cell cycle arrest; tumour cell death	154 ± 22.19	0.232	−23.7 ± 2.71	7.5 mg/kg twice, one week apart	Reduced tumour growth	[163]
Stroke										
	RVG29-PLGA-PEG NP	Rat model of cerebral ischemia	Baicalin	Neuroprotection	120	0.18	−3	9 mg/mL 3 days before modeling	Reduced neuroinflammation	[182]
Epilepsy										
	PLGA NP	Pentylene- tetrazole-induced seizure rat model	Lamotrigine	Reduces neuronal excitation to suppress seizure activity	170 ± 2.8	0.191 ± 0.035	−16.6 ± 2.96	0.833 mg/kg 15 min before induction of seizure activity	Delayed seizure onset	[164]
	Chitosan-PLGA NP	Pentylene- tetrazole- and increasing current electroshock-induced seizure rat model	Catechin hydrate	Antioxidant, anti-inflammatory properties	93.46 ± 3.94	0.106 ± 0.01	−12.63 ± 0.08	10 mg/kg	Increased seizure latency and threshold	[170]
Depression										
	Chitosan-PLGA NP	Stress and reserpine induced rat model	Desvenlafaxine succinate	Inhibition of serotonin and noradrenaline re-uptake	172.5 ± 10.2	0.254	35.63 ± 8.25	5 mg/kg daily in 50 μL per nostril for 16 days	Reduced symptoms of depression, increased levels of serotonin, noradrenaline and dopamine	[161]

PDI; polydisperity index, GBM; glioblastoma multiforme, SLN; solid lipid nanoparticle, PLGA NP; poly(lactic-co-glycolic acid) nanoparticle, PEG; polyethylene glycol, DA; dopamine, WGA; wheat germ agglutinin, EphA3; ephrin type-A receptor 3, NMDA; N-methyl-D-aspartate, AChE; acetylcholinesterase, 6-OHDA; 6-hydroxydopamine, MPTP; 1-methyl-4-phenyl-1,2,3,6-tetrahydropyridine, L-Dopa; levodopa, VEGF; vascular endothelial growth factor, RVG29; peptide isolated from the rabies virus to facilitate NP brain uptake.

## 6. Conclusions and Future Perspectives

CNS diseases are typically accompanied by life-long prescriptions. However, many current therapies are inadequate by means of efficacy or tolerability, or both. The BBB severely limits the ability of drugs to enter the brain and new strategies are desperately needed to improve drug efficacy and associated adverse effects. Nanomedicine, particularly polymeric and lipid-based NPs, can be loaded with drugs, proteins, peptides, and gene therapies for targeted brain delivery of treatments for CNS diseases. Furthermore, intranasal delivery of drug loaded nanoparticles can bypass the BBB and be formulated as self-administering dosages, preventing patient compliance issues. 

The physicochemical properties of PLGA NPs and SLNs make them suitable for CNS drug delivery; they are biocompatible and cross the BBB in vitro and in vivo. This review highlights the ability of PLGA and SLNs to deliver therapeutics to the CNS and provide therapeutic relief in animal models of a variety of brain diseases. Intranasal delivery of PLGA NPs and SLNs encapsulating drugs bypass the BBB, further improving brain uptake and reducing systemic drug actions that cause adverse effects. 

The evidence that nose-to-brain transport of drug-loaded PLGA NPs and SLNs is efficacious in animal models of CNS disease is in the early stages, but research suggests that this strategy is optimal for increased brain concentrations of the drug, reduced peripheral drug accumulation, and disease-modifying outcomes. Further investigations are required to confirm the advantages of intranasal polymeric and lipid-based drug administration prior to the inevitable translation to the clinic.

## Figures and Tables

**Figure 1 pharmaceutics-15-00746-f001:**
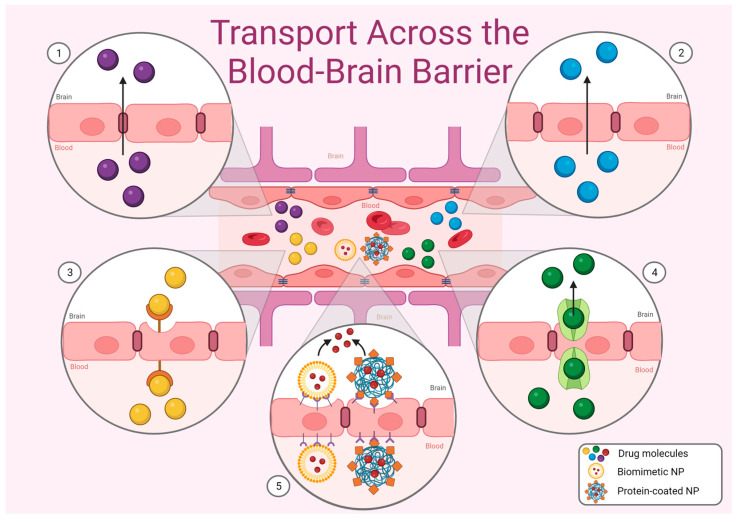
Transport across the BBB primarily occurs by paracellular transport (1), passive diffusion (2), receptor-mediated transcytosis (3) or carrier-mediated transport (4). Nanoparticles (NPs) can also cross the BBB for CNS drug delivery (5); biomimetic NPs synthesized using physiological proteins, cell membranes and viruses take advantage of the natural uptake of these materials. Additionally, synthetic nanoparticles can be coated in targeting ligands such as transferrin, P-glycoprotein and angiopep-2 that bind to receptors located on BBB cells to facilitate permeation of the BBB and drug release within the brain parenchyma (Created in BioRender.com; accessed on 2 February 2023).

**Figure 2 pharmaceutics-15-00746-f002:**
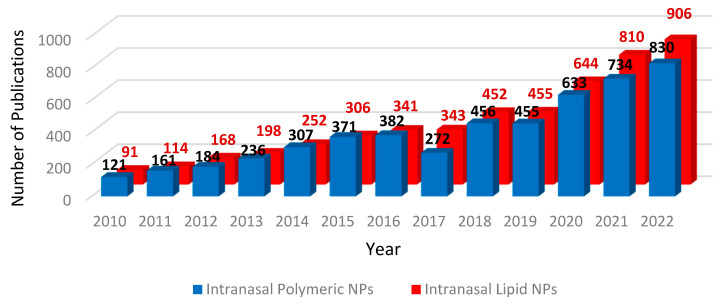
Advances in intranasal polymeric and lipid-based nanocarriers in the past decade. Searches were carried out in the Scopus database using search terms “intranasal” and “polymer” and “nanoparticle” or “intranasal” and “lipid” and “nanoparticle”.

**Figure 3 pharmaceutics-15-00746-f003:**
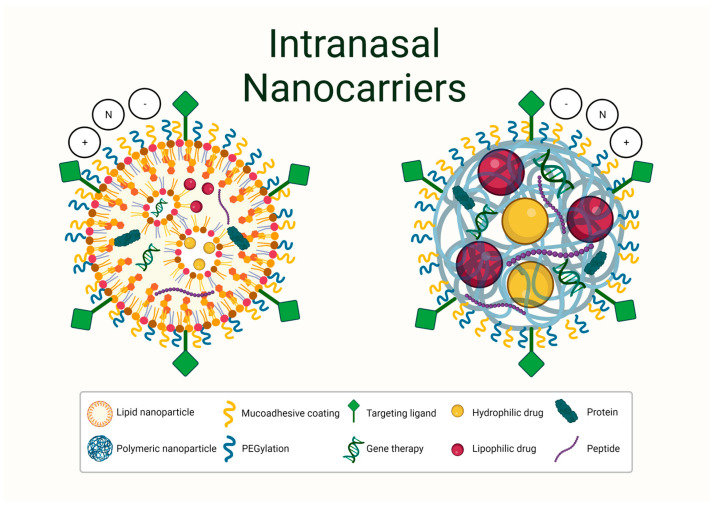
Both hydrophilic and lipophilic drugs as well as gene therapies, proteins and peptides can be encapsulated by SLNs and PLGA NPs. Both particle types can be conjugated with poly(ethylene glycol) (PEG) or saccharides to improve biocompatibility and/or targeting proteins or antibodies that bind to receptors located on endothelial cells of the BBB or within the brain parenchyma to enhance CNS uptake. SLNs and PLGA NPs can be synthesized with different surface charges. For intranasal delivery of NPs, a mucoadhesive coating is often used to enhance residence time in the nasal cavity which facilitates nose-to-brain transport (Created in BioRender.com; accessed on 2 February 2023).

**Figure 4 pharmaceutics-15-00746-f004:**
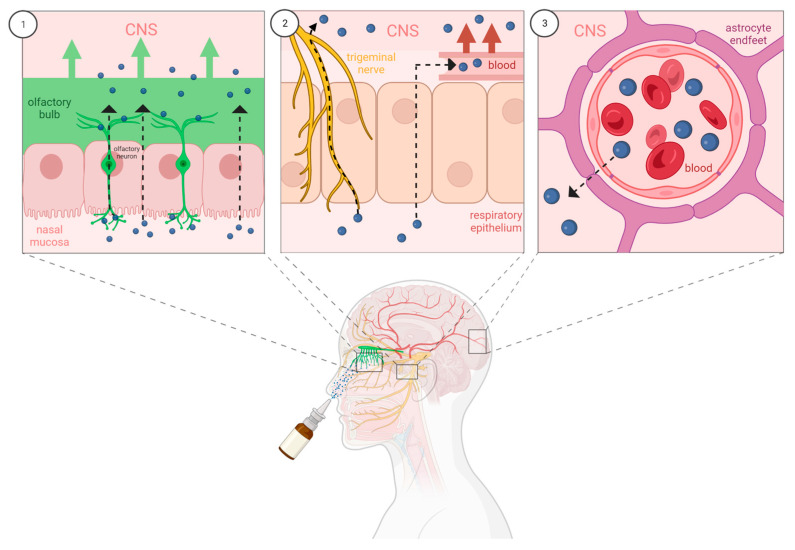
(1) Direct transport of intranasal drugs from the nasal mucosa to the olfactory bulb can occur by axonal transport along olfactory neurons or para- or transcellular transport across the nasal epithelium. (2) In the respiratory region of the nasal cavity, drugs can be endocytosed by the trigeminal nerve and travel along the axon to reach the CNS or can cross epithelial cells to reach the blood. (3) Once in the blood, drugs administered intranasally must cross the BBB to reach the CNS [108] (Created in BioRender.com; accessed on 2 February 2023).

## Data Availability

No new data were created or analyzed in this study. Data sharing is not applicable to this article.

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
