# Peer review of "Intranasal Polymeric and Lipid-Based Nanocarriers for CNS Drug Delivery"

_pharmaceutics, 2023, doi:10.3390/pharmaceutics15030746_

Round 1

Reviewer 1 Report

The review article titled “Intranasal Polymeric and Lipid-based Nanocarriers for CNS Drug Delivery” by Maher et al. has been reviewed where the authors explored the use of polymeric and lipid-based nanocarriers for drug delivery and their potential in repurposing drugs to treat CNS diseases. The delivery method focused in this article is nasal way. The article is well prepared has covered the recent publications in this field. The failure to achieve FDA approval of recently developed drugs has been discussed in the light of their poor delivery during the clinical trial. This is an interesting and informative article. In my opinion addressing some of the following concerns will further improve it.

11. Is it possible to show the recent growth of this field through a plot showing the number publications each year? This plot can present two aspects: (i) number of articles published each year during the past decade where polymeric nanocarriers was used for intranasal pathway, and (ii) number of articles published each year during the past decade where lipid-based nanocarriers was used for intranasal pathway. Then the progress in both the areas will be understandable and easy to compare. From this observation, the reasons for the variation in progress of these two nanocarrier-based delivery systems can be explained which if added to the manuscript will provide an important aspect of this study. Figure 14 of the reference Advanced Science, 2022, 9, 2105373 (DOI: 10.1002/advs.202105373) can be an example for this. This section can be added towards the beginning of the introduction or a best suitable place of the manuscript.

22. Line 111: Would the authors consider using 1-999 nm instead of 1-1000 nm?

33. Section 2.1: Although PLGA is known as biodegradable polymer, however few recent reports have mentioned about its toxicity. Hence more studies are required on toxicity of such polymers (cf. above mentioned reference which also has other references therein related to this aspect). I think this information can be included as many review articles are missing this.

44. Adding a section on the influence of surface charges of these nanoparticles on the BBB permeation seems missing. Not including this is like missing an important aspect which has significant influence on the efficacy of BBB permeation, cellular uptake, and drug delivery. Although some data in the table is provided, general explanation about the influence of surface charge and a figure to explain this would be helpful to easily understand the influence of cationic, anionic, and neutral charges.

55. Section 2.6.2.: Is there any study where these nanocarriers are used for treating ‘diffuse intrinsic pontine glioma (DIPG)’ of children?

66. Section 41.: Can the mechanism of nasal delivery bypassing the BBB be presented through a schematic figure to make it easy for the readers to follow? J. Pharm. Investigation, 2023, 53, 119-152, DOI: 10.1007/s40005-022-00589-5 and similar references might be helpful. In the figure please out how and where the BBB is getting bypassed.

Author Response

The authors thank the editors and review panel for their time and insightful feedback on the review titled ‘Intranasal Polymeric and Lipid-based Nanocarriers for CNS Drug Delivery’. Based on your valuable comments, improvements have been made to this manuscript (see tracked changes).

Point-by-point responses to each comment and how they have been addressed can be found below in red text. The authors hope that these amendments are satisfactory but welcome any further suggestions for this manuscript.

Reviewer 1:

  1. “Is it possible to show the recent growth of this field through a plot showing the number publications each year? This plot can present two aspects: (i) number of articles published each year during the past decade where polymeric nanocarriers was used for intranasal pathway, and (ii) number of articles published each year during the past decade where lipid-based nanocarriers was used for intranasal pathway. Then the progress in both the areas will be understandable and easy to compare. From this observation, the reasons for the variation in progress of these two nanocarrier-based delivery systems can be explained which if added to the manuscript will provide an important aspect of this study. Figure 14 of the reference Advanced Science, 2022, 9, 2105373 (DOI: 10.1002/advs.202105373) can be an example for this. This section can be added towards the beginning of the introduction or a best suitable place of the manuscript.”

Response: The authors appreciate this suggestion and thank the reviewer for directing to this helpful resource. We have represented the growth of the field in Fig. 2 with key search terms including “intranasal” and “polymer” and “nanoparticle” or “intranasal” and “lipid” and “nanoparticle”. This has also been addressed in the preceding text; “Over the past decade, the growth of the field of both intranasal polymeric and lipid-based nanocarriers as CNS drug delivery devices has accelerated, reflected by the numbers of publications (figure 2).”

  1. “Line 111: Would the authors consider using 1-999 nm instead of 1-1000 nm?”

Response: The authors have now used 1-999 nm instead of 1-1000nm.

  1. “Section 2.1: Although PLGA is known as biodegradable polymer, however few recent reports have mentioned about its toxicity. Hence more studies are required on toxicity of such polymers (cf. above mentioned reference which also has other references therein related to this aspect). I think this information can be included as many review articles are missing this.”

Response: The authors thank the reviewer for their insightful suggestion and agree that such information would be a beneficial addition to the manuscript. Hence, the following has been included in the text (section 2.1): “Increases in PLGA NP size and concentration as well as changes in shape have been reported to cause cytotoxicity in vitro, resulting in macrophage activation and the production of reactive oxygen species (ROS). Nevertheless, the body of evidence suggesting that PLGA is biocompatible far exceeds those that describe toxicity and so, further studies are required to investigate physiological and toxicological responses to PLGA in vivo.”

  1. “Adding a section on the influence of surface charges of these nanoparticles on the BBB permeation seems missing. Not including this is like missing an important aspect which has significant influence on the efficacy of BBB permeation, cellular uptake, and drug delivery. Although some data in the table is provided, general explanation about the influence of surface charge and a figure to explain this would be helpful to easily understand the influence of cationic, anionic, and neutral charges.”

Response: The authors thank the reviewer for this feedback and agree that such information is a crucial element to the review. Hence, a section on the influence of surface charge has been added to the manuscript (section 2.3).

  1. “Section 7.2.: Is there any study where these nanocarriers are used for treating ‘diffuse intrinsic pontine glioma (DIPG)’ of children?”

Response: The authors have made reference to a recent review of nanoparticles for the diagnosis and treatment of paediatric brain cancers (doi:10.3390/diagnostics12010173) in section 2.7.2.

  1. “Section 4.1.: Can the mechanism of nasal delivery bypassing the BBB be presented through a schematic figure to make it easy for the readers to follow? J. Pharm. Investigation, 2023, 53, 119-152, DOI: 10.1007/s40005-022-00589-5 and similar references might be helpful. In the figure please show how and where the BBB is getting bypassed.”

Response: We thank reviewer 1 for this helpful suggestion and for providing an example. Fig. 4 has been updated to include the mechanism of nasal delivery allowing for bypassing the BBB.

Reviewer 2 Report

In this review, the authors tried to summarize the recent progress of drug delivery systems from intranasal route to CNS. Basically, this review is helpful to readers in understanding the basic knowledge and state-of-art of this area. There are several issues that should be addressed as shown below:

1, Biomimetic nanoparticles prepared from endogenic species that help cross BBB possible via the receptor-mediated transport, for instance, 10.1016/j.apsb.2017.09.008, maybe could be supplemented under Fig. 1.

2, The title is mainly focusing intranasal CNS delivery, however, the authors introduce too much content in BBB crossing, until Page 12.

3, There are various materials for NP besides PLGA. The authors are suggested to introduce more species.

4, More applying examples should be included, with schemes or illustrations, with the aim of showing the detailed design for CNS drug delivery from nose-to-brain route.

5, The design principle for NP delivering to brain intranasally should be summarized.

6, Only neurodegenerative disease and brain tumors were exhibited in this review as disease models, which should be extended.

Author Response

The authors thank the editors and review panel for their time and insightful feedback on the review titled ‘Intranasal Polymeric and Lipid-based Nanocarriers for CNS Drug Delivery’. Based on your valuable comments, improvements have been made to this manuscript (see tracked changes).

Point-by-point responses to each comment and how they have been addressed can be found below in red text. The authors hope that these amendments are satisfactory but welcome any further suggestions for this manuscript.

Reviewer 2

  1. “Biomimetic nanoparticles prepared from endogenic species that help cross BBB possible via the receptor-mediated transport, for instance, 10.1016/j.apsb.2017.09.008, maybe could be supplemented under Fig. 1.”

Response: We thank reviewer 2 for this suggestion and have amended Fig. 1 to include the mechanism of BBB uptake for biomimetic NPs and NPs coated with targeting ligands via receptor-mediated transport.

  1. “The title is mainly focusing intranasal CNS delivery, however, the authors introduce too much content in BBB crossing, until Page 12.”

Response: We thank reviewer 2 for highlighting this issue. To address this point, information on the surface coating of NPs for intranasal delivery has been added to section 2.4. and early content on other administration routes has been better linked to intranasal delivery. As the intranasal delivery of particles is a developing field, there are limited examples of the efficacy of drugs in CNS disease models, with most studies thus far focusing on elements of particle design or the biodistribution of particles in the brain following intranasal delivery in healthy animals. Therefore, the prior content, particularly in section 3.1. and 3.2 is necessary to indicate that both PLGA NPs and SLNs do not affect various organs and tissues in the body and that drug delivery via these particles has the potential to be disease modifying in a range of CNS disorders. Based on this feedback, linking section 3.1. and 3.2. to later sections and highlighting the relevance of studies conducted via alternative routes has better integrated the information to focus on intranasal CNS delivery.

  1. “There are various materials for NP besides PLGA. The authors are suggested to introduce more species.”

Response: We thank reviewer 2 for this comment. There are indeed many polymers that can be used to synthesise NPs besides PLGA. Examples of these materials are listed in section 2.1. The authors decided to focus on one class of polymer and lipid-based NP for this review as the scope was too broad and the manuscript too long when trying to include multiple types of each. Following an extensive literature search, the authors concluded that PLGA NPs and SLNs were the most common type of polymer and lipid-based NPs that were synthesised with the intent of CNS drug delivery and had been investigated in animal models of CNS disease. We also would like to draw attention to sections 2.1. and 2.2., where the reasons for focusing on PLGA NPs and SLNs are highlighted.

  1. “More applying examples should be included, with schemes or illustrations, with the aim of showing the detailed design for CNS drug delivery from nose-to-brain route.”

Response: We thank reviewer 2 for this suggestion and have updated Fig.3 to include the design principles for intranasal delivery. This comment  and response overlaps with comment 6 from reviewer 1. Based on this feedback from both reviewers, Fig. 4 has been amended to highlight the transport route of NPs from nose-to brain.  

  1. “The design principle for NP delivering to brain intranasally should be summarized.”

Response: The authors thank reviewer 2 for this recommendation. We agree that this is a critical piece of information and have included the following in the abstract; “NPs can be specifically designed for intranasal administration by tailoring their size and coating with mucoadhesive agents or other moieties that promote transport across the nasal mucosa.” These design characteristics are further detailed in Fig. 3 with supporting information. Additionally, in section 5.2. the following text has been added “Based on the evidence presented here, for the successful delivery of CNS drugs to the brain the following should be considered in the design of intranasal NPs; the use of biocompatible and lipophilic materials for NP synthesis, particle size, surface charge and mucoadhesion.”

  1. “Only neurodegenerative disease and brain tumours were exhibited in this review as disease models, which should be extended.”

Response: We thank reviewer 2 for this feedback. We assume that this comment refers to section 5.2 and the accompanying table 2. While many studies of NP biodistribution and drug bioavailability of intranasal PLGA NPs and SLNs are available (described in section 5.1.), there are limited pharmacodynamic studies on the effect of intranasal drug delivery via PLGA NPs and SLNs on CNS disease in animal models. Despite extensive literature searches, the studies listed in table 2 which were conducted in animal models of Alzheimer’s disease, Parkinson’s disease, glioblastoma, epilepsy and depression were the only evidence reported to date of nose-to-brain drug delivery via PLGA or SLN nanocarriers in preclinical CNS disease models. Since receiving the reviewer’s feedback, a further literature search has resulted in the addition of a similar study in stroke but to date, intranasal PLGA NPs and SLNs have not been investigated in animal models of CNS disorders like traumatic brain injury, schizophrenia and anxiety, to the best of the author’s knowledge.

Round 2

Reviewer 1 Report

The authors have made sufficient effort to improve their manuscript following the comments by reviewer. I expect that this article will attract many readers and I am glad to reccomend it for publication in Pharmaceutics.